# A Coordinated Response at The Transcriptome and Interactome Level is Required to Ensure Uropathogenic *Escherichia coli* Survival during Bacteremia

**DOI:** 10.3390/microorganisms7090292

**Published:** 2019-08-25

**Authors:** Natalia Sanchez de Groot, Marc Torrent Burgas

**Affiliations:** 1Gene Function and Evolution Lab, Centre for Genomic Regulation (CRG), Dr. Aiguader 88, 08003 Barcelona, Spain; 2Systems Biology of Infection Lab, Biochemistry and Molecular Biology Department, Universitat Autònoma de Barcelona, 08193 Cerdanyola del Vallès, Spain

**Keywords:** *Escherichia coli*, urinary tract infection, bacteremia, antimicrobials

## Abstract

Localized infections or disruption of the skin barrier can enable the entry of bacteria into the bloodstream, possibly leading to acute inflammation and sepsis. There is currently no holistic view on how bacteria can survive and spread in the bloodstream. In this context, we combined transposon mutagenesis, gene-expression profiling and a protein interaction network analysis to examine how uropathogenic *Escherichia coli* can proliferate in blood. Our results indicate that, upon migration from the urea to serum, *E. coli* reacts to the osmolarity difference, triggering a transcriptomic response in order to express survival genes. The proteins codified by these genes are precisely organized at the interactome level and specifically target short linear motifs located in disordered regions of host proteins. Such a coordinated response helps to explain how bacteria can adapt to and survive environmental changes within the host. Overall, our results provide a general framework for the study of bacteremia and reveal new targets for potential study as novel antimicrobials.

## 1. Introduction

Urinary tract infection (UTI) is one of the most common bacterial infections in adults and one of the most severe bacterial infections in infants [1]. UTIs affect more women than men and approximately half of women will, at least, develop one episode of UTI during their lifetime [2]. Most worryingly, UTIs frequently relapse and account for up to 10–20% of antibiotic prescriptions in ambulatory care, representing a focus of resistance emergence [3,4,5]

UTIs proceed in an ascending manner in the genitourinary tract, beginning with bacteria infecting the bladder and later ascension in the kidneys, where they can cause pyelonephritis [6]. In some cases, bacteria can cross the epithelial cells of the proximal tube and endothelial cells of the capillaries to enter the bloodstream, initiating bacteremia that can later become sepsis [7]. *Escherichia coli* is the most common etiologic agent, accounting for almost 75% of all Gram-negative bacteremias, with uropatogenic *E. coli* (UPEC) being responsible for more than 80% of uncomplicated UTIs [8]. UPEC is classified as extraintestinal pathogenic *E. coli*. Although most UPEC strains lack a type III secretion system [9], they do express an arsenal of virulence factors that allow tissue infection.

Usually, UPEC infections are treatable with antibiotics, typically trimethoprim/sulfamethoxazole or ciprofloxacin, but recent reports [10] highlight an increase in antibiotic resistance that, together with an alteration of the gut microbiota, contribute to treatment failure. Accordingly, expanding our knowledge of UPEC virulence factors and how they interact with the host could contribute to the design of new therapies for the treatment of resistant strains. In the present study, we examine the proteins that contribute to the fitness of the CFT073 infection, which is a well-known UPEC strain. We demonstrate that, during the transition from urine to blood, *E. coli* cells express efflux pumps and proteins related to the antimicrobial peptide defense. Most interestingly, these proteins are essential for survival in blood and show a high degree of modularity. Our results represent an integrated framework with which to understand how bacterial cells respond to environmental changes and may help in the design of new and specific treatments for bacteremia.

Despite recent advances in the field of genomics and proteomics, we still lack an integrated view on how bacteria and their hosts respond in the context of infection. More specifically, we do not completely understand how bacteria react to different host niches during infection. In this context, we integrated different datasets to provide a general framework to analyze the impact of gene expression in bacteremia. Using CFT073 as a model, we set out to answer three basic questions: (1) What are the genes that have a specific role in bacteremia and how are they regulated at the transcriptional level? (2) How do these proteins interact with each other to promote a coordinated response? (3) What are the molecular determinants of these interactions?

## 2. Materials and Methods 

### 2.1. Databases

The *E. coli* CFT073 fitness dataset was obtained from [11]. Briefly, a super-saturating pool of 360,000 independent transposon mutants in a CFT073 strain cultured from the blood of a patient with pyelonephritis was used to infect 6- to 7-week-old CBA/J mice. Transposon insertion sites before (input) and after (output) infection were defined using transposon directed insertion-site sequencing (TraDIS). Fitness values were calculated as the ratio of population expansion for the two genotypes using the following equation:(1)$$F^{a}=\frac{C_{i}^{a}/N_{i}}{C_{0}^{a}/N_{0}}$$
where $F^{a}$ is the fitness factor for gene *a*, $C_{i}^{a}$, $C_{0}^{a}$ are the number of reads of gene *a* before and after infection and $N_{i}$, $N_{0}$ are the number of total reads before and after infection, respectively.

Transcriptome analyses were obtained from the following microarray datasets: [12] (serum); [13] (urine); and [14] (NaCl and urea). In the serum-related datasets, *E. coli* CFT073, grown in a mid-exponential phase, was incubated in either LB or LB supplemented with 50% human serum. For the urine-related datasets, *E. coli* CFT073 was grown to the mid-exponential phase in either LB or filtered human urine. Finally, NaCl and urea data were obtained by determining the transcriptional adaptive responses of the UPEC strain CFT073 to the presence of 0.3 M NaCl or 0.6 M urea in LB. The *E. coli* CFT073 interactome was obtained from the String database; only experimentally validated interactions were included. The *H. sapiens- E. coli* interactome was obtained by a homology search using the HPIDB database. The *E. coli* CFT073 genome was obtained from the NCBI (CFT073 accession number NC_004431).

### 2.2. Functional and Genome Enrichment Analyses

Functional enrichment analyses were used to identify groups of genes that were over-represented in the fitness dataset. These groups of genes were enriched in one or several features, suggesting an association with the disease. Analyses were conducted using DAVID (Database for Annotation, Visualization, and Integrated Discovery [15] (http://david.abcc.ncifcrf.gov/). Enrichment was considered to be significant when the adjusted *p*-value was < 0.05 using a Benjamini-Hochberg correction, unless otherwise specified. WoPPER was used for genome enrichment, with fitness scores used as the input dataset. WoPPER [16] uses the input data to perform a local adaptive smoothing of the data over the gene coordinates. Local maxima and minima were estimated for the smoothed profile against the expected null distribution and clusters were identified.

### 2.3. Disorder Propensity and Short Linear Motif Calculations

A large fraction of our proteome is composed of proteins that lack globular structure. These proteins have, in whole or in part, an undefined three-dimensional structure. Unstructured proteins represent hubs in the human interactome and are often associated with diseases. For this reason, it is important to study the role of these proteins in relation to human diseases. The analysis of disorder propensity and the search of putative linear motifs was undertaken using the IUPred [17] (long mode; Available online: http://iupred.enzim.hu/) and ANCHOR [18] (Available online: http://anchor.enzim.hu/) algorithms, respectively.

### 2.4. Calculation of Network Parameters

Proteins generally do not act alone but interact with each other to perform their functions. Protein network analyses focus primarily on the interrelationships between proteins, rather than on the proteins themselves. Hence, such networks can be used to discover protein functions and disease associations. Protein networks were analyzed using R [19] and Cytoscape [20]. The degree (*k*) of a node *n* is defined as the number of edges linked to *n*. Degree distributions were fitted to a power law distribution defined as P(k)_~_k^−γ^, where γ is the degree exponent. The clustering coefficient (C_n_) of a node *n* with degree *k* was calculated as *C_n_* = 2*e_n_*/(*k_n_*(*k_n_*−1)), where *k_n_* is the degree of *n* and *e_n_* the number of connected pairs between all the neighbors of *n*.

Clusters were identified using the Cluster One application for Cytoscape [21]. Fitness values were used as edge weights and the remaining parameters were set to default values.

### 2.5. Statistical Analyses

Unless otherwise specified, all the *p*-values were calculated using the Mann-Whitney U-test and were considered to be significant when *p* < 0.05. 

## 3. Results and Discussion

### 3.1. Characterisation of Bacteremia Fitness-Related Genes in CFT073 Infection

The fitness of a gene can be defined as the contribution it makes to the global genotype of subsequent generations. For infectious organisms, such fitness unavoidably measures the contribution of the gene to the infection, as only those bacteria able to survive the host response will proliferate. To measure gene fitness during CFT073 bacteremia, we used previously published transposon mutagenesis data [11] to determine the contribution of individual genes to bacteremia. In such experiments, a well-characterized collection of transposon mutants was sequenced before and after injection into 6-7-week-old CBA/J mice (Figure 1a). The fitness value for a given gene (F^a^) was calculated as the proportion between the relative number of reads before and after the injection (See Materials and Methods; Figure 1a). We first noticed that fitness values were not normally distributed, but instead highly skewed, with most genes contributing little or nothing to bacteria fitness (Figure 1b). We categorized CFT073 genes in three groups according to their quartile position (F1, F2 and F3; Figure 1b). Genes belonging to the first quartile were labelled F1. For these genes, the relative number of reads increased, on average, four to five times (fitness values ≥ 4; Figure 1b, red shading). Genes belonging to the second quartile were labelled as F2 genes and had a moderate impact on fitness, increasing more than twice but less than four times the relative number of reads (4 > fitness values ≥ 2; Figure 1b, orange shading). Finally, the F3 group contained genes that belong to the third and fourth quartiles and have a very limited or null effect on bacteria fitness (2 > fitness values ≥ 0; Figure 1b, grey shading).

We performed pathway enrichment analysis in the KEGG database to investigate the functions of the genes belonging to the categories defined above. Our analysis revealed distinct enrichment features (Figure 1c). The F1 genes were particularly enriched in two-component system genes, and also ABC transporters and flagella (*p*-value < 10^−6^, using a Benjamini-Hochberg test). Proteins related to flagellar assembly and two-component systems in the F1 genes included multidrug efflux, lipopolysaccharide modification, cell wall and membrane composition, type III secretion systems, and the antimicrobial peptide response (Appendix A). F2 and F3 genes were mainly enriched in ABC transporters and other pathways related to metabolism, e.g., amino acid or purine metabolism (Appendix A). Overall, these results suggest that, during bacteremia, bacteria must be prepared to resist a specific antimicrobial response in serum, including effectors secreted by innate immune cells such as antimicrobial peptides. Bacteria thus need to modify their cell wall structure and use efflux pumps to counteract host defenses. Although these genes are required to develop bacteremia in mice, it is unclear whether they are specifically transcribed in response to the host. Accordingly, by activating these specific functions, we investigated whether these genes are regulated at the transcriptional level during bacteremia. 

### 3.2. Differential Expression of CFT073 Bacteremia Fitness-Related Genes

As UPEC initiates infection in the urinary tract and must migrate to the kidney before reaching the bloodstream, we examined the transcriptional response of uropathogenic *E. coli* CFT073 in the presence of both serum and urine, using previously published datasets. When comparing the three previously defined gene groups (F1, F2 and F3), we observed that expression levels are dependent on the fitness group (Figure 1d, *p*-value = 0.0031 using the Kruskal-Wallis test). In addition, the mean expression value for genes belonging to F1 was significantly higher than the value for genes in the F3 group (Figure 1d, *p*-value = 0.034). This observation suggests the existence of a transcriptional program triggered upon contact with human serum.

Surprisingly, we observed an opposite trend when comparing expression data from urine. In that case, the mean expression value for genes belonging to F1 was significantly lower compared to F3 (Figure 1d, *p*-value = 0.0108). One possible explanation is that the osmolarity difference between urine (high concentration of urea and sodium chloride) and serum (absence of urea and moderate concentration of sodium chloride) may trigger transcriptional reprogramming. To test this hypothesis, we explored the expression of the gene groups F1, F2 and F3 in the presence of urea and sodium chloride. We observed that under higher osmotic pressure, the mean expression value for genes belonging to F1 was lower than the value for F3 genes (Figure 1d, *p*-values 2.32·10^−5^ and 0.046 for NaCl and urea, respectively). 

Lastly, we also explored which genes display the highest expression difference between serum and urine, i.e., which genes are highly up-regulated in urine but down-regulated in serum and vice versa (Figure 1e). Some genes showed a characteristic rank increase (Figure 1e, orange genes) or decrease (Figure 1e, red genes). Among the orange genes, we detected enrichment (*p* < 0.05 using a Fisher Exact test) for two-component systems, e.g., arcAB and baeRS, which are important in the resistance to oxygen-reactive species and antimicrobials, respectively. Among the red genes, most belong to sugar transport and metabolism (e.g., the galactose and maltose transporters mglA and malG, respectively).

Overall, these data suggest that bacteria may use differences in osmolarity between urine and blood to activate transcriptional reprogramming in order to trigger the expression of virulence and survival factors.

### 3.3. Genome organisation of CFT073 Bacteremia Fitness-Related Genes

As we found a transcriptional reprogramming of *E. coli* when moving from urine to serum that is consistent with the expression patterns of bacteremia, we analyzed the genome distribution of the F1, F2 and F3 genes. We observed that the F1 genes are mostly packed in a limited region of the genome, with maximum density at 2 Mb (Figure 2a). In contrast, the F2 and F3 genes are more evenly distributed along the genome. Given that F1 genes were more condensed along the genome, we investigated whether we could identify clusters of genes with high average fitness values. To this end, we used Locally Adaptive Procedure (LAP) algorithms to detect groups of physically contiguous genes characterized by similar profiles [16]. We were able to detect two clusters in the aforementioned region that contain genes with a high impact on fitness. Unfortunately, no enrichment could be detected in these clusters, because most of the proteins (>60%) correspond to hypothetical or uncharacterized proteins (Appendix A). Although this obviously hinders our capacity to identify relevant genes. we could identify some that were related to stress-protection, such as thiol peroxidase and iron and murein membrane-associated proteins. Our results remind us again of the importance of gene annotation. Much work is still required in this field, as > 50% of genes are classified as hypothetical or uncharacterized, even in the well-characterized *E. coli* K-12 [22]. 

### 3.4. Interactome Connectivity of Fitness-Related Genes

Although F1 genes are clustered in the genome and their transcription is triggered after they reach the bloodstream, it is important to identify how the proteins codified by these genes interact in order to understand how they perform functions in a coordinated manner. To this end, we analyzed the *E. coli* interactome to understand how proteins in each group interact with each other, how they are organized and how such organization is related to function. To quantify this, we reconstructed the protein-protein interactome for all subsets of genes (F1, F2 and F3) using the STRING database (the F1 interactome is displayed in Figure 3a).

First, we observed that the network density (a measure to calculate how the network is populated by edges) for proteins belonging to F1 is higher than for the other groups (F2 and F3, Figure 3b). This is also true for the average path length in the network that measures the expected distance between two nodes and how fast the information can travel in the network (Figure 3b). These results suggest that proteins that are most relevant for bacteremia (i.e., those having a high impact on fitness) are more interconnected than the remaining genes. When analyzing the properties of the F1 network, we observed that the topology of the network is compatible with small-world networks (Figure 3c). This network also shows a moderate degree of hierarchy, which makes it possible to identify modules in the network (Figure 3c). These modules are also associated with functions relevant to bacteremia (Figure 3d): peptide transport is intimately related to antimicrobial peptide resistance; flagellar assembly is activated for infection of host cells; siderophore expression is crucial in the uptake of iron; and phosphotransference is associated with changes in protein regulation and may be important when it comes to activating the cascades required for survival. Moreover, metabolic changes must occur, as the bacteria are growing in a very different environment.

Overall, our results suggest that, during bacteremia, the increase in the expression of fitness-related genes can potentially rewire the bacteria interactome to perform the required functions for bacteria survival.

### 3.5. Coordination of the Host-Pathogen Interactome 

Some of these proteins not only interact with other bacterial proteins but can also be secreted into the host cell. Indeed, several reports have shown that secreted pathogen proteins can rewire the host interactome to hijack the host cell machinery to facilitate infection [23]. Accordingly, as well as evaluating the pathogen interactome, it is also important to investigate how pathogen proteins can interact with the human interactome.

To this end, we reconstructed the CFT073-human interactome using the HPIDB database, searching for all possible host-pathogen interactions based on sequence homology in the known interactions. The resulting interactome is depicted in Figure 4a (details of all interactions are available in the Appendix A). Although interactomes predicted by homology may be incomplete, we estimate that the coverage is sufficient for statistical purposes given the information stored in the dataset (~50,000 interactions) and the diversity of the organisms (65 host and 652 pathogen species). We observed that human proteins targeted by *E. coli* have a higher disorder propensity than the average human proteome (Figure 4b, based on IUPred prediction). We also observed that these human proteins have, on average, more short linear motifs (SLiMs) than the average human proteome (Figure 4c, based on ANCHOR prediction). These features are appealing for two reasons: 1) SLiMs found in disordered regions are generally regulatory regions and so control central processes in the cell such as phosphorylation [24]; and (2) SLiMs, especially if they are found in disordered regions, are easy to model and can be attractive when it comes to the design of new drugs [25].

To go even deeper, we determined which SLiMs are enriched in human-targeted proteins compared to other proteins in the human genome (Figure 4d). To this end, we used ANCHOR to compute all the SLiMs in all the human proteins and then plotted the enrichment based on how many times a given motif was present in the set of human proteins targeted by *E. coli* compared to the average human proteome (Figure 4d). Setting a fairly strict two-fold threshold, we found that four SLiMs were overrepresented in the targeted set compared to the human proteome. These SLiMs were all found preferentially in disordered regions of the proteins (Figure 4e) and are involved in pathways that are especially relevant for infection. Among the most represented motifs, we found SH3 domains that are involved in signal transduction pathways, cytoskeleton organization, membrane traffic or organelle assembly. These motifs play a fundamental role in host invasion. Similarly, EVH and PXL motifs are related to endosomal sorting and cytoskeleton organization at the membrane level, respectively. Finally, TRAF motifs are related to adaptor proteins associated with TNF receptors.

Overall, these results indicate that proteins secreted by *E. coli* to the host cell milieu can interact with non-self host proteins and interfere with the normal functioning of the cell. Based on our results, we propose that proteins can use SLiMs located in disordered regions to hijack the cell functioning to help bacteria to infect and survive inside the host.

## 4. Conclusions

Our results suggest that, to reach the bloodstream from the kidney, UPEC *E. coli* CFT073 sense the difference in osmolarity and initiate a reprogramming of the transcriptional profile. Some up-regulated genes are involved in a large number of interactions and coordinate a remodeling of the protein interactome that enhances survival in the new environment, including the control of the antimicrobial response by the host defenses. Concomitantly, virulence factors are secreted to modulate the interaction with the host. More specifically, secreted virulence factors interact with proteins and regulators in the host cell, allowing the pathogen to rewire the network of interactions [9,21]. These factors are specialists in interacting with SLiMs displayed by host proteins in disorder-prone segments.

The results obtained here are consistent with other results reported by other *E. coli* strains. For example, several ABC transporters identified by network analysis in the present study (e.g., IroN, TonB), are required in most *E. coli* strains causing urinary tract infections and gastroenteritis [22]. Two component systems and flagellar components are also required for infection and highly conserved among *E. coli* strains and also closely related to other species such as *Shigella* [23,24].

A thorough study of fitness-related genes may produce successful candidates for drug discovery. Our results suggest that protein-protein interactions, both in the host and between the host and the pathogen, may be interesting targets for antibiotic development. Especially interesting are the interactions between host and pathogen proteins that involve linear segments. These regions may be more easily modelled using linear peptides and small drugs as new resource antibiotics and may provide a new class of drugs for the emerging problem of bacterial resistance to classical antibiotics. Further investigations of how bacteria sense osmolarity differences may lead to the development of potential drugs for bacteremia. By blocking certain triggers in this process, we may be able to block the transcription of virulence genes that allow bacteria to be cleared from the bloodstream by the immune system.

## Figures and Tables

**Figure 1 microorganisms-07-00292-f001:**
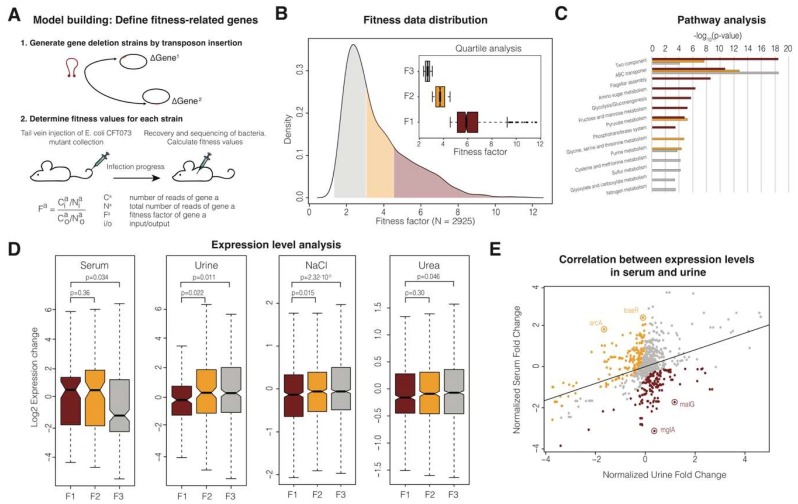
Schematics of the experimental design, fitness definition and gene expression analysis. (**A**) To determine the fitness of each gene in the *E. coli* CFT073 genome, transposons are inserted at random in the genome, generating a pool of mutants that can be characterized by sequencing. To determine the contribution of each gene to the fitness of the organism, this mutant pool can be injected in mice and any remaining bacteria recovered and sequenced after infection. Those mutants unable to infect the host will be depleted from the output pool compared to the input pool (fitness values > 1). (**B**) Density plot showing the distribution of the calculated fitness values. Genes were classified according to quartiles; F1 genes correspond to the first quartile (dark red), F2 genes to the third quartile (orange) and grey genes to both the first and second quartiles (grey). The distribution of fitness values for each gene group is depicted as a boxplot in the insert. (**C**) Pathway analysis enrichment for each gene group. *p*-values are represented for each group and pathway (only significant pathways with an adjusted *p*-value < 0.05 using the Benjamini-Hochberg correction are shown in the figure). (**D**) Expression levels for genes in four different conditions: serum, urine, sodium chloride (NaCl) and urea. Statistical analyses were performed using the Mann-Whitney U-test and *p*-values corresponding to all the comparisons are shown in the figure. The median value for each group of proteins is shown with a horizontal black line. Boxes represent values between the first and third quartiles. All the values outside this range are considered to be outliers and were removed to improve visualization. The notches correspond to a ~95% confidence interval for the median. (**E**) Correlation of gene expression levels between serum and urine. Expression levels were first normalized using z-scores. Differentially expressed genes between serum and urine are highlighted in orange (>two-fold rank increase in serum compared to urine) and red (> two-fold rank increase in urine compared to serum). Some exemplars in both sets are highlighted in the figure.

**Figure 2 microorganisms-07-00292-f002:**
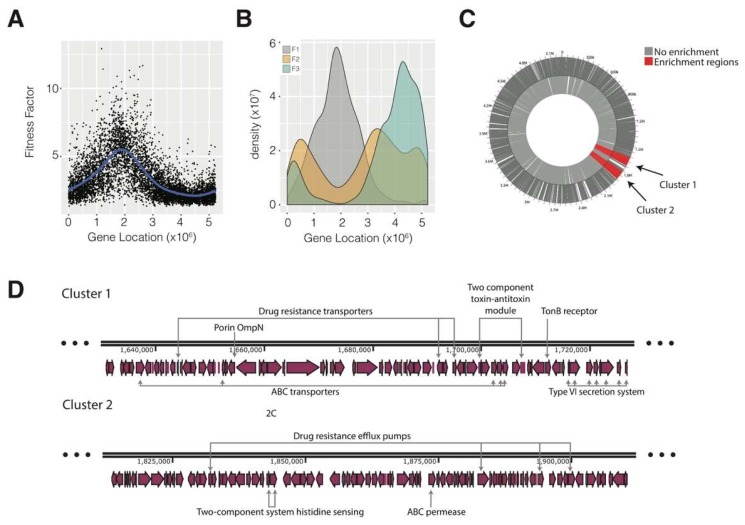
Gene localization analysis and fitness enrichment in the *E. coli* genome. (**A**) Distribution of fitness factors along the *E. coli* CFT073 genome (**B**) Genome localization of genes according to the fitness classification (F1 genes in grey, F2 genes in orange and F3 genes in green). (**C**) Positional enrichment analysis of fitness-related genes in the genome. According to WoPPER, two clusters of genes were identified. The functional enrichment of both clusters was weak due to a high number of uncharacterized and hypothetical proteins. (**D**) Detail of the two clusters identified by positional enrichment. Some proteins belonging to the F1 main functional categories were identified in both clusters and are highlighted as exemplars in the figure, e.g., two-component system and ABC transporters.

**Figure 3 microorganisms-07-00292-f003:**
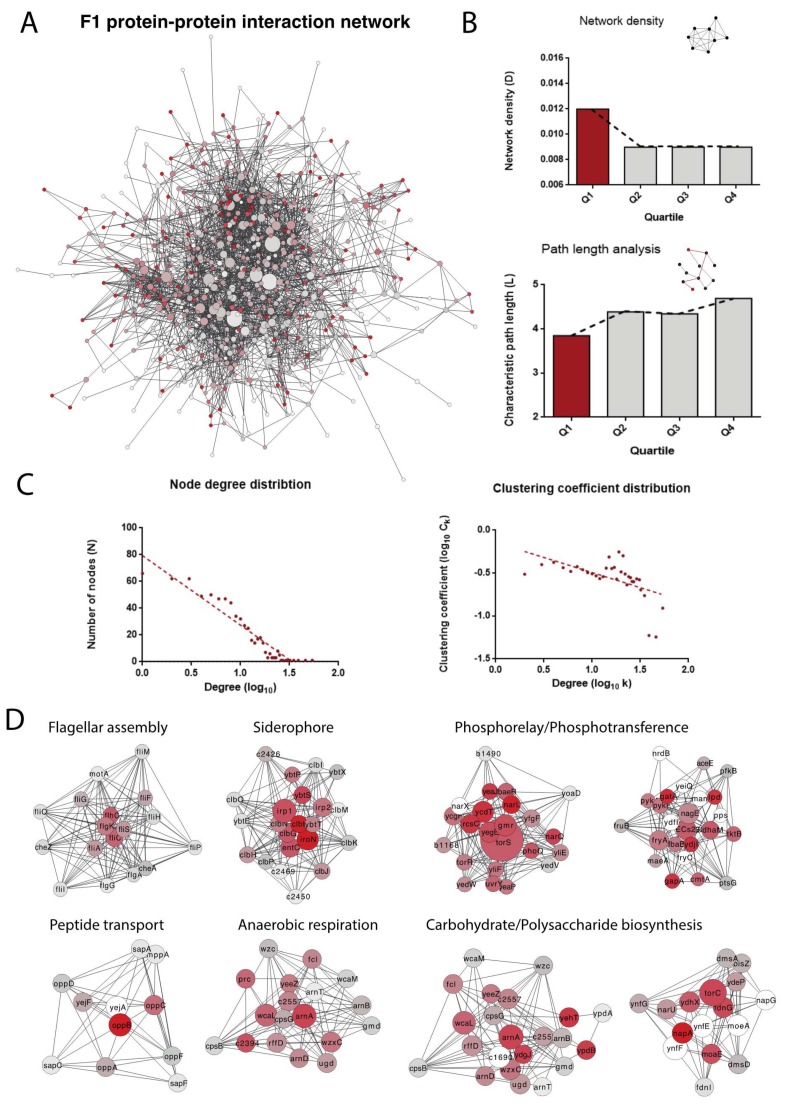
Protein-protein interconnectivity of fitness-related genes. (**A**) Representation of the protein-protein interaction network of genes belonging to the F1 group, based on data deposited in the String database. Only experimentally validated interactions were used. The size of the nodes in the network indicates betweenness centrality, i.e., the number of shortest paths that pass through the node. Node size is proportional to betweenness centrality, colour from grey to red indicates an increasing clustering coefficient and line width is proportional to edge centrality. (**B**) Analysis of the network density and pathway length for the F1, F2 and F3 networks. (**C**) Node degree and clustering coefficient distribution of the F1 interaction network. The plots show that the F1 network follows a typical free-scale distribution with a moderate degree of modularity. (**D**) Protein clusters detected for the F1 network using the Cluster One algorithm. Only clusters with *p*-values < 0.05 were selected (*p*-values were calculated using the Mann-Whitney U test performed on the in- and out-weights of the vertices).

**Figure 4 microorganisms-07-00292-f004:**
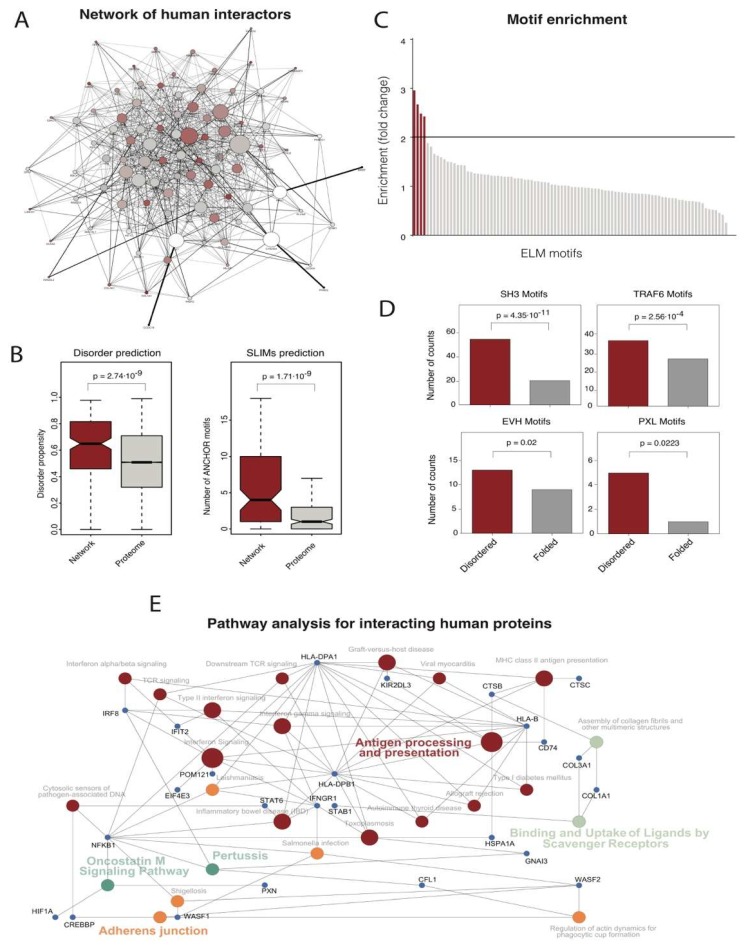
*Characteristics of human protein targeted by E. coli.* (**A**) Representation of the *E. coli-H. sapiens* protein-protein interaction network of genes belonging to the F1 group based on data deposited in the String database. (**B**) Distributions of disorder propensity and the number of ANCHOR motifs in human proteins targeted by *E. coli* CFT073 F1 proteins compared to the human proteome. (**C**) Enrichment analysis of ELM motifs in human proteins targeted by *E. coli* CFT073 F1 proteins compared to the human proteome. The number of ELM signatures for all the human proteins was calculated, as was the ratio between the observed motifs in the human proteins targeted by the *E. coli* CFT073 F1 proteins and the human proteome counts. Motifs with > two-fold enrichment are colored in dark red. (**D**) Distribution of enriched motifs in human proteins targeted by *E. coli* CFT073 F1 proteins between folded and disordered regions. In all cases, statistical analyses were performed using the Mann-Whitney U-test and *p*-values corresponding to all the comparisons are shown in the figure. The median value for each group of proteins is shown with a horizontal black line. Boxes represent values between the first and third quartiles. All values outside this range are considered to be outliers and were removed to improve visualization. The notches correspond to a ~95% confidence interval for the median. (**E**) Pathway enrichment analysis (colored circles) for human proteins targeted by *E. coli* CFT073 proteins. Most relevant proteins connecting all pathways are shown as blue circles and labelled with the protein name.

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
