# Peer review of "A Coordinated Response at The Transcriptome and Interactome Level is Required to Ensure Uropathogenic Escherichia coli Survival during Bacteremia"

_microorganisms, 2019, doi:10.3390/microorganisms7090292_

Round 1

Reviewer 1 Report

The manuscript entitled „A coordinated response at the transcriptome and interactome level is required to ensure uropathogenic Escherichia coli survival during bacteremia” is a very interesting paper, which uses current methodology to answer the research questions. The results and discussion are adequately presented, and the manuscript holds relevant information.

I have several comments before this manuscript should be considered for publication:

I would reconsider the number and nature of the designated keywords for this paper. Lines 26-31 should not be the introductory section of the paper, instead, start with the relevance of UTIs in human infections, their clinical relevance and then incorporate their role in bacteremia. “Gram-negative” is the correct way to reference these bacteria. E. coli should always be in italics (both in the text of the MS and in the References). Line 36: I suggest the citation of the following articles regarding UTIs (especially regarding pyelonephritis):

Medicina 2019, 55(6), 285; https://doi.org/10.3390/medicina55060285

Antibiotics 2019, 8(3), 91; https://doi.org/10.3390/antibiotics8030091

Line 40: I recommend the citation of the following article, regarding UPEC:

Medicina 2019, 55(7), 356; https://www.ncbi.nlm.nih.gov/pubmed/26443763

Line 47: I suggest the addition of the following citation:

Molecules 2017, 22(3), 468; https://www.ncbi.nlm.nih.gov/pmc/articles/PMC4790395/

The authors should elaborate on the relevance and real-word value of “Functional and genome enrichment analyses”, “Disorder propensity and short linear motif calculations” and “Calculation of network parameters”, because this is not currently available in the manuscript, and this may lead to confusion and a narrower readership for this paper. The importance of these methods should be described in more detail (either in the M&M or R&D sections), in addition, they should be explained in such a way, that even people unfamiliar with these calculations realize WHY these methods were used during data analyses. The Figures should be larger and may be a bit more segmented (more individual figures, instead of A-X sections on each figure) because the bulk of the information is presented on these. Additionally, some of these figures lack in quality.

If the authors adhere to these revisions, I fully recommend the acceptance of this article.

Reviewer 2 Report

Dear Authors,

The manuscript ID: microorganisms-576021-v1 entitled “A coordinated response at the transcriptome and interactome level is required to ensure uropathogenic Escherichia coli survival during bacteremia” written by Natalia Sanchez de Groot and Marc Torrent Burgas is very interesting. This article (as the Authors summarized) provide a general framework for the study of bacteremia and reveal new targets for potential study as novel antimicrobials.

The Authors performed difficult, comprehensive and detailed research related to the survival of bacteria Escherichia coli during bacteremia. Manuscript is very well written and organized. The methodology is properly described. The results are accurately presented in the form of figures, appropriately interpreted and discussed. Statistical analyzes were also used. Conclusions are justified.

I have no negative comments for this paper, except for a very small suggestion in order to improve it, namely: in the text and figures – please write E. coli in italics.

Best regards,
